# Rhizosphere Microbiota Promotes the Growth of Soybeans in a Saline–Alkali Environment under Plastic Film Mulching

**DOI:** 10.3390/plants12091889

**Published:** 2023-05-05

**Authors:** Han-Cheng Mao, Yifei Sun, Chengyuan Tao, Xuhui Deng, Xu Xu, Zhenquan Shen, Laijie Zhang, Zehui Zheng, Yanhua Huang, Yongren Hao, Guoan Zhou, Shulin Liu, Rong Li, Kai Guo, Zhixi Tian, Qirong Shen

**Affiliations:** 1Jiangsu Provincial Key Lab of Solid Organic Waste Utilization, Jiangsu Collaborative Innovation Center of Solid Organic Wastes, Educational Ministry Engineering Center of Resource-Saving Fertilizers, Nanjing Agricultural University, Nanjing 210095, China; 2Laboratory of Bio-Interaction and Crop Health, Nanjing Agricultural University, Nanjing 210095, China; 3Institute of Environment and Ecology, Shandong Normal University, No. 88, Wenhuadong Road, Lixia District, Ji’nan 250014, China; 4Institute of Genetics and Developmental Biology, Chinese Academy of Sciences, Beijing 100101, China

**Keywords:** rhizosphere microbes, plastic film mulching, saline–alkali soil, soybean

## Abstract

The rhizosphere microbiota plays a critical and crucial role in plant health and growth, assisting plants in resisting adverse stresses, including soil salinity. Plastic film mulching is an important method to adjust soil properties and improve crop yield, especially in saline–alkali soil. However, it remains unclear whether and to what extent the association between these improvements and rhizosphere microbiota exists. Here, from a field survey and a greenhouse mesocosm experiment, we found that mulching plastic films on saline–alkali soil can promote the growth of soybeans in the field. Results of the greenhouse experiment showed that soybeans grew better in unsterilized saline–alkali soil than in sterilized saline–alkali soil under plastic film mulching. By detecting the variations in soil properties and analyzing the high-throughput sequencing data, we found that with the effect of film mulching, soil moisture content was effectively maintained, soil salinity was obviously reduced, and rhizosphere bacterial and fungal communities were significantly changed. Ulteriorly, correlation analysis methods were applied. The optimization of soil properties ameliorated the survival conditions of soil microbes and promoted the increase in relative abundance of potential beneficial microorganisms, contributing to the growth of soybeans. Furthermore, the classification of potential key rhizosphere microbial OTUs were identified. In summary, our study suggests the important influence of soil properties as drivers on the alteration of rhizosphere microbial communities and indicates the important role of rhizosphere microbiota in promoting plant performance in saline–alkali soil under plastic film mulching.

## 1. Introduction

Saline-alkali soil is widespread worldwide and is regarded as one of the major abiotic factors that adversely influence plant growth and production worldwide [1,2]. It has been estimated that more than 1000 million hectares of soil are threatened by salinization and alkalinization accounting for 7.5% of the world’s total area [3,4]. Even worse, the figure is growing at a rate of more than 10% per year [5]. Extensive studies have demonstrated that soil salinization and alkalization induce osmotic and ionic stresses, hinder nutrient cycling, and accelerate plant senescence [6,7,8], particularly in agricultural ecosystems. This leads to a reduction in crop yield and causes massive amounts of economic losses every year [9]. Nevertheless, the increasing fertility of these soils after desalinization [10,11] emphasizes the utilizability of saline–alkali soils. Thus, ameliorating saline–alkali soils to enhance plant growth is conducive to the healthy sustainable development of agriculture.

Plastic film mulching (PM) technology is a low-cost, easily operable and high-durability agricultural practice for raising crop production [12] that has been widely adopted around the world, particularly in Europe, Africa, Asia and America [13]. In the last two decades, PM has led to a 25% to 42% increase in global crop yield on average [14], and the application of plastic films on arable lands has increased tremendously in recent years, attaining over 1.5 million tons annually worldwide, especially in Chinese croplands. The proportion of PM has reached 60–80% [15,16].

It is widely reported that PM can increase the crop yield. Although the potential of PM in conserving soil moisture [17], modifying soil temperature [18], and reducing weed pressure [19] are often considered to be the main reasons for the crop harvests in previous studies, the influence of PM on soil microbiota cannot be ignored [20,21]. Plant-associated microbiota, the second genome of plants, are crucial for plant health [22]. Root-derived microbiomes play an integral role in this complex community [23,24], including a class of microbes that inhabit the narrow zone of roots known as the rhizosphere [25]. Mounting evidence signifies the importance of rhizosphere microorganisms in enhancing plant growth [26,27], suppressing soil-borne disease [28,29], and improving the capability to fight against abiotic stresses such as drought [30], cold [31] and salt [32,33,34]. However, the functional link between PM-induced shifts in rhizosphere microbial communities and crop yield remains essentially unknown.

In this study, the rhizosphere microbes were considered. We hypothesized that soybean yield enhancement would be realized in both soil physicochemical properties and rhizosphere soil microbiota, with special key taxa representing the microbial group most responsive to PM and contributing to soybean yield enhancement. Through a field survey and a greenhouse mesocosm experiment, we found that PM and soil microbiota were indeed important factors in promoting soybean growth in saline-alkali land. To further understand the role of rhizosphere microbiota in yield enhancement in saline–alkali soil under PM practices, we studied the soil physicochemical properties and rhizosphere soil microbiota, with a particular emphasis on the linking of them. Afterward, we pinpointed 11 rhizosphere bacterial and three rhizosphere fungal OTUs that had been reported to have potential salt-tolerant and growth-promoting functions in previous studies. The aims of the study were to (1) interpret the important role of rhizosphere microbes in the process of increased yield by PM according to the observations from the field survey and greenhouse experiment, and (2) identify the specific classifications of the potential key rhizosphere bacteria and fungi associated with grow-promoting effects. This study may provide a reference for the development of microbial agents.

## 2. Results

### 2.1. Soybean Yield and Bulk Soil Properties in the Field Survey

In all cultivars, the CF treatment significantly (Student’s *t* test, *p* < 0.01) increased soybean yield (43.66% higher) (Figure 1a).

For bulk soil properties, the moisture content in the CF treatment was significantly higher (Student’s *t* test, *p* < 0.01) than that in the C treatment (4.95% higher) (Figure 2a). The electrical conductivity (EC) in the CF treatment was significantly lower (Student’s *t* test, *p* < 0.01) than that in the C treatment (27.41% lower) (Figure 2b). We observed a similar phenomenon regarding K^+^ and Na^+^ (25.78% and 32.85% lower, respectively) (Student’s *t* test, *p* < 0.001 and Student’s *t* test, *p* < 0.001, respectively) (Figure 2c,d). Details of soybean yield (Appendix A) and bulk soil properties (Appendix A) are provided in the Appendix A.

### 2.2. Results of the Greenhouse Mesocosm Experiment

To investigate the influence of plastic film mulching and rhizosphere soil microbes on the growth of soybeans planted in a salty environment, a greenhouse experiment containing four treatments (US + F = unsterilized soil with plastic film mulching, US + NF = unsterilized soil without plastic film mulching, S + F = sterilized soil with plastic film mulching, and S + NF = sterilized soil without plastic film mulching) was performed.

We found that the US + F treatment performed best and the S + NF treatment performed the worst in promoting the growth of soybeans (Figure 1(b1)). S + F had a better plant growth promotion effect (Student’s *t* test, *p* < 0.05) than S + NF, US + F exhibited a significantly (Student’s *t* test, *p* < 0.05) stronger growth when challenged by salinity in comparison with S + F, and US + F also showed a better effect than US + NF (Student’s *t* test, *p* < 0.01) (Figure 1(b1)). However, no significant difference was observed between US + NF and S + NF (Student’s *t* test, *p* > 0.05).

Variance partitioning analysis (VPA) reflected the contributions of films and soil microbiota (56% and 10% explanation, respectively) to the growth of soybeans (Figure 1(b2)). Two-way ANOVA drew a similar conclusion (Appendix A).

### 2.3. Diversity and Structure of Soybean Rhizosphere Microbiota

The bacterial Shannon index in the C and CF treatments was similar (Student’s *t* test, *p* > 0.05) to each other (Figure 3a), as was that for fungi (Student’s *t* test, *p* > 0.05) (Figure 3b). Principal coordinate analysis (PCoA) related to Bray-Curtis distance was conducted to distinguish the differences in community structures of rhizosphere bacteria and fungi across the two treatments. Results showed that the C treatment distinctly separated from the CF treatment for both bacteria (Figure 3c and Appendix A, PERMANOVA, *p* < 0.01) and fungi (Figure 3d and Appendix A, PERMANOVA, *p* < 0.01). A random forest model showed that fungal PCoA1 and bacterial PCoA2 had significantly (*p* < 0.001 and *p* < 0.05, respectively) the highest and second-highest MSE% values in their contributions to soybean yield, while bacterial Shannon and fungal Shannon were not significant (Figure 3e).

The abundance of total rhizosphere bacteria and fungi in the CF treatment was significantly (Student’s *t* test, *p* < 0.05) higher, but not significant higher (Student’s *t* test, *p* > 0.05), than that in the C treatment, respectively (Appendix A). The rhizosphere microbial community compositions between the C and CF treatments were detected at the phylum level. For bacteria, Proteobacteria, Bacteroidetes, Actinobacteria and Chloroflexi were present at a higher relative abundance than other phyla in both treatments (Appendix A). For fungi, Ascomycota and Basidiomycota had the highest and second-highest relative abundance, respectively (Appendix A).

### 2.4. Key Microbial Taxa Associated with Yield and Soil Characteristics

To evaluate the effect of plastic film mulching on the communities of rhizosphere microbes, linear discriminant analysis of effect size (LEfSe) was applied at the OTU level. In general, 95 bacterial and 65 fungal OTUs in the rhizosphere differed under the influence of plastic film mulching. Two sets of those 95 bacterial and 65 fungal OTUs were defined as differential bacterial and fungal OTUs, respectively. For bacteria, 44 OTUs were enriched in the CF treatment, of which 38 OTUs showed a 2-fold significant increase in relative abundance (|log_2_ fold change of OTUs| > 1, *p* < 0.05) compared to the C treatment (Figure 4a). For fungi, 32 OTUs were enriched in the CF treatment, of which the relative abundance of 14 OTUs were significantly higher by 2-fold (|log_2_ fold change of OTUs| > 1, *p* < 0.05) compared to C (Figure 4b).

Then, the significance of differential bacterial and fungal OTUs linked to soybean yield was calculated using a random forest model. In different bacterial taxa, 23 OTUs were observed to have significant (*p* < 0.05) and positive contributions to soybean yield, and OTU3058 was the highest factor explaining 10.03% of the yield increase (Appendix A). For fungal taxa, 11 OTUs had a significant (*p* < 0.05) and positive influence on soybean yield, and OTU1537 had the highest explanation, 12.15% (Appendix A).

Further analysis with Spearman’s rank correlation revealed that 11 OTUs had significant and positive correlations (FDR adjusted *p* < 0.05, R > 0) with yield among the 23 bacterial OTUs (Table 1 and Figure 4c). For fungi, the number was three among 11 OTUs (Table 1 and Figure 4c). These 11 and three OTUs were defined as potential key rhizosphere bacterial and fungal OTUs, respectively.

A heatmap was plotted to show the association between the alterations of bulk soil characteristics under plastic film mulching and the variations of the relative abundance of potential key rhizosphere microbial taxa (Figure 5). All 14 OTUs were positively correlated with moisture content but negatively correlated with EC, K^+^ and Na^+^, including 11 bacterial genera (*Methylovorus*, *Oharaeibacter*, *Pseudarthrobacter*, *Neobacillus*, *Peribacillus*, *Acinetobacter*, *Olivibacter*, *Stenotrophomonas*, *Neorhizobium*, *Dethiosulfatarculus*, and *Methylophilus*) and three fungal (*Atractiellales_unidentified_1*, *Pyrigemmula*, and *Fusarium*). In addition, they were only enriched in the rhizosphere soil with plastic film mulching and were 2-fold significantly higher in relative abundance (log_2_ fold change of OTUs > 1, *p* < 0.05) compared with the C treatment, except OTU1525.

The conceptual model (Figure 6) shows that the optimization of saline-alkali soil properties initiated by plastic film mulching indirectly encourages the growth of soybeans by driving the increase in the relative abundance of potential key rhizosphere microbes that are active in the rhizocompartment. Furthermore, it is an undeniable fact is that the mitigation of environmental stress eases the survival pressure of plants and can promote the growth of soybeans directly.

## 3. Discussion

In this study, we learned from a field survey and a greenhouse experiment that plastic film mulching, and soil microbiota have a growth-promoting effect on soybeans under saline-alkali stress. We further analyzed the differences in rhizosphere bacterial and fungal communities between the two treatments and explained the reason for these differences on the growth promotion of soybeans by linking the alterations of bulk soil characteristics after plastic film mulching. Additionally, we pinpointed the classifications of potential key rhizosphere microbes.

Plant growth promotion is inseparable from complex and diverse shifts in rhizosphere microbial communities and is closely related to the presence and activity of specific microbial populations [35,36,37], i.e., potential key rhizosphere microbial taxa. It is possible to understand the important role played by these special rhizosphere microbial taxa in the life of plants through integrating and analyzing the sequencing data of rhizosphere microbiota [38]. In our study, the significantly higher soybean yield in the CF treatment than in the C treatment observed from the field survey and significantly sturdier soybean growth in the S + F treatment than in the S + NF treatment observed from the greenhouse experiment are supported by previous studies showing that plastic film mulching could help plants grow better [18,20]. The significantly better soybean growth in the US + F treatment than in the S + F treatment observed from the greenhouse experiment is consistent with previous findings indicating that the rhizosphere microbiota could help plants grow better [22,39]. However, no significant difference was detected between the US + NF treatment and S + NF treatment. This is possibly because the environment of saline-alkali soil without plastic film mulching is detrimental to the survival and activity of soil microbiota, especially the functional rhizosphere microbial taxa. Furthermore, the results of variance partitioning analysis (VPA) and two-way ANOVA also revealed the separate and direct effects of plastic film mulching and soil microbiota on promoting the growth of soybeans under saline-alkali stress (Figure 1(b2) and Appendix A).

We further analyzed the high-throughput sequencing data of rhizosphere soil bacteria and fungi after establishing the importance of soil microbiota in soybean growth. The results showed that rhizosphere bacterial and fungal PCoA contributed significantly and positively to the soybean yield, rather than the Shannon (Figure 3e). No significant differences were detected in the Shannon of rhizosphere bacteria and fungi between the two treatments, but rhizosphere bacterial and fungal PCoA were markedly separated. These results imply that the community structures (beta diversity) of rhizosphere bacteria and fungi were changed under plastic film mulching, and these changes could affect soybean yield, which can be supported by previous discoveries [17]. Furthermore, the results of linear discriminant analysis of effect size (LEfSe) showed that these changes mainly reflected in the significant differences in the taxa and abundance of soybean rhizosphere microbes enriched in the two treatments. Specifically, 95 bacterial and 65 fungal OTUs were distinguished significantly between the two treatments (Figure 4a,b). Some of these OTUs that may promote the growth of soybeans, i.e., potential key rhizosphere microbial taxa, were detected notably higher abundance (|log_2_ fold change of OTUs| > 1, *p* < 0.05) and were 100% enriched in the CF treatment compared with the C treatment, and soybean yield was significantly and positively correlated with these OTUs. These results further imply a linkage between higher soybean yield in the saline–alkali environment and a higher relative abundance of potential key rhizosphere bacterial and fungal taxa.

Why were potential key rhizosphere microbes enriched in the CF treatment? In previous studies, researchers pointed out that plant rhizosphere microbiota were related to the soil environment [37,40], and soil with different properties could shape diverse rhizosphere microbiota [41]. The significantly higher moisture content and lower EC, Na^+^, and K^+^ detected in the CF bulk soil samples are in line with previous findings showing that plastic film mulching could reduce soil salinity and maintain soil moisture [42]. This is probably because plastic films can reliquefy evaporated water back into the soil to keep it moist and make soil salinity percolate into the groundwater zone by leaching, thus reducing soil salt content and improving soil quality [43]. Conversely, in fields without plastic film mulching, soil salt tends to shift upward by evaporation, resulting in the accumulation of salinity in the root zone, while the salt concentration tends to increase [44]. Higher water content and lower salinity detected in the saline–alkali lands provide optimized conditions for soil and a more appropriate rhizocompartment for the survival and proliferation of potential key rhizosphere microbes [45,46,47], resulting in a significantly higher relative abundance of them in the CF treatment. Therefore, the correlations between potential key rhizosphere microbiota and soil moisture or salinity targets (EC, Na^+^, and K^+^) were found to be positive or negative (Figure 5), respectively.

The potential key rhizosphere microbial taxa identified at the genus level included *Methylovorus*, *Methylophilus*, *Ohareaibacter*, *Pseudarthrobacter*, *Neobacillus*, *Peribacillus*, *Acinetobacter*, *Stenotrophomonas*, *Neorhizobium*, *Olivibacter*, and *Dethiosulfatarculus* for bacteria and *Atractiellales_unidentified_1*, *Fusarium*, and *Pyrigemmula* for fungi. Some of them have been reported in previous studies. *Methylovorus* (OTU3058) and *Methylophilus* (OTU3336), belonging to the family Methylophilaceae, were demonstrated to have a high tolerance to salinity stress [48] and could accelerate the growth of tobacco, potato, and tomato [49,50,51]. *Ohareaibacter* (OTU3280), belonging to the order Rhizobiales, could tolerate 0 to 2% NaCl [52], and its potential in growth-promoting on leguminous plants was suggested by the ability of the encoding of nitrogenase [53]. *Pseudarthrobacter* (OTU2779) can synthesize plant hormones, such as indole acetic acid (IAA), indicating agricultural potential [54,55]. *Neobacillus* (OTU3480) and *Peribacillus* (OTU369) were recently proposed as genera separated from *Bacillus* [56]. *Peribacillus* could facilitate the seed germination and growth of maize in a saline environment [57]. *Neobacillus* has mostly been reported as a root endophytic bacterium [58,59] that can promote plant growth [60]. *Acinetobacter* (OTU113) was found to promote the growth of pearl millet seedlings in pot experiments [61]. *Stenotrophomonas* (OTU3424) is a kind of PGPB (plant growth-promoting bacteria) that can produce IAA in vitro [62], tolerate high osmotic conditions (highest 5% NaCl), grow at 4 °C to 40 °C [63], and subsequently influence plant growth. *Neorhizobium* (OTU125) has the characteristics of salt tolerance (highest 4.5% NaCl), alkali stability (highest pH 9.5) and low temperature resistance (lowest 10) °C [64], and it was proven to increase the biomass of vegetables [65,66] due to the production of siderophores and the encouragement of IAA-generating bacteria in the rhizosphere environment [64]. Moreover, the nitrogen-fixing gene indicates its importance in promoting the growth of legumes [67]. However, no literature records are found for the potential importance of *Olivibacter* (OTU313) and *Dethiosulfatarculus* (OTU935) in plant growth promotion. For fungi, an unidentified genus (OTU1537) belonging to the order Atractiellales was also detected in soybean roots in previous studies [68], and its function in improving plant growth based on mechanisms to be studied was confirmed in inoculation trials [68]. The genus *Fusarium* (OTU1525) was found to produce a positive return on soybean growth in our system. Interestingly, strains belonging to the genus *Fusarium*, such as *F. solani* and *F. oxysporum*, are usually regarded as well-known pathogens causing diseases in different crops [69,70]. However, researchers have also pointed out the possibility of *Fusarium* promoting plant growth in some specific cases, e.g., ecological competition for nutrients results in pathogens exhibiting a disease-suppressive effect [29], and nonpathogenic *Fusarium* strains promote plant growth directly by producing secondary metabolites and inducing resistance [71,72,73]. However, the role of *Pyrigemmula* (OTU113) in plant growth remains unreported.

## 4. Materials and Methods

### 4.1. Root and Soil Samplings

During autumn and winter of 2020, a field survey was conducted to collect root samples of soybeans associated with 12 cultivars (Appendix A) growing in a natural saline–alkali environment in Dongying (37°46′ N, 118°49′ E), Shandong Province, China. Two different treatments were set up with these soybean plants: C = chemical fertilizer, and CF = chemical fertilizer with plastic film mulching.

Bulk soil samples were collected simultaneously according to the method described by Li et al. [5]. In brief, we removed the 0–10 cm layer and collected the 10–25 cm layer. Appendix A shows the selected characteristics of the bulk soil samples. Moisture content was calculated as dry soil weight (105 ℃ for 12 h) divided by fresh soil weight. Electrical conductivity (EC) was determined by using an EC meter, and K^+^ and Na^+^ levels were determined by flame photometry.

Rhizosphere soil samples were collected immediately according to Tao et al. [74] after the roots were brought back to the lab in a fresh-preserved container. Briefly, the root samples were shaken to remove bulk soil and rinsed with sterile stroke–physiological saline solution. Then, the supernatant of the mixture was separated using a centrifuged at 10,000× *g* for 10 min, and the precipitate defined as rhizosphere soil. All rhizosphere soil samples were stored at −80 °C for DNA extraction.

### 4.2. DNA Extraction, Sequencing and qPCR

All rhizosphere soil samples were prepared for 16S rRNA gene sequencing. According to the manufacturer’s guide of the DNeasy^®^ PowerSoil^®^ Kit (QIAGEN, Inc., Redwood City, CA, USA), microbial DNA was extracted from 0.5 g rhizosphere soil. A NanoDrop 2000 spectrophotometer (Thermo Scientific, Waltham, MA, USA) was used to determine the concentration and quality of the DNA samples.

Bacterial and fungal sequencing libraries were constructed according to the MiSeq Reagent Kit Preparation Guide (Illumina, San Diego, CA, USA) as described previously [75]. The paired-end amplicons were sequenced on an Illumina MiSeq platform (Illumina, San Diego, CA, USA) according to standard protocols at Magigene Technology Co., Ltd. (Guangdong, China). The forward primer 515F (5′-GTGCCAGCMGCCGCGGTAA-3′) and the reverse primer 907R (5′-CCGTCAATTCMTTTRAGTTT-3′) were used to amplify the V4-V5 hypervariable region of the bacterial 16S rRNA gene. For targeting the ITS1-1 region of the fungal ITS gene, the forward primer ITS5-1737F (5′-GGAAGTAAAAGTCGTAACAAGG-3′) and the reverse primer ITS2-2043R (5′-GCTGCGTTCTTCATCGATGC-3′) were used.

The copy number of total rhizosphere bacteria and fungi in the two treatments was quantified using qPCR. The primers were Eub338 (5′-ACTCCTACGGGAGGCAGCAG-3′)/Eub518 (5′-ATTACCGCGGCTGCTGG-3′) and ITS1f (5′-TCCGTAGGTGAACCTGCGG-3′)/5.8 s (5′-CGCTGCGTTCTTCATCG-3′) for bacteria and fungi, respectively. The reaction system contained 10 µL SYBR Green premix Ex Taq (2×), 0.5 µL each primer, 8 µL double-distilled water, and 1 µL template DNA. The thermal cycling processes were 95 °C for 30 s, 95 °C for 5 s with 40 cycles and 65 °C for 34 s.

### 4.3. Greenhouse Mesocosm Experiment

In spring 2021, a greenhouse experiment was performed in an Intelligent Greenhouse at Nanjing Agricultural University, Nanjing, Jiangsu Province, China. The saline–alkali soil used was gathered from Yancheng (33°31′ N, 120°10′ E), Jiangsu Province, China, using the method described by Niu et al. [76]. The properties of the experimental soil (Appendix A) were determined. The soybean cultivar used in this study was TZX-805, povided by the Institute of Genetics and Developmental Biology, Chinese Academy of Sciences.

The greenhouse experiment consisted of four treatments: US + F = unsterilized saline–alkali soil with plastic film mulching, US + NF = unsterilized saline–alkali soil without plastic film mulching, S + F = sterilized saline–alkali soil (75 kGy γ-irradiation) with plastic film mulching, and S + NF = sterilized saline–alkali soil (75 kGy γ-irradiation) without plastic film mulching. All soybean seeds were surface sterilized following the method described by Li et al. [5], with some modifications. Briefly, the seeds were soaked in 75% ethanol for 1 min and 0.3% sodium hypochlorite for 30 s, and then rinsed 5 times with sterile distilled water. Then, three seeds were sown in each plot containing 1 kg of the above saline–alkali soil, and all plots were placed under greenhouse conditions (16 h photoperiod, 20 °C/28 °C, night/daytime). The seedlings were thinned to one for each pot after they developed two true leaves, and the plants were irrigated with sterile distilled water as needed [77]. The physiological properties of the plants were determined after 28 days of growth.

### 4.4. Bioinformatics and Statistical Analysis

The quality of raw 16S rRNA (bacteria) and ITS (fungi) sequences was checked by FastQC [78] and performed using USEARCH as described by Zhang et al. [79]. To generate the OTU table, the qualified reads were clustered into operational taxonomic units (OTUs) with the similarity of 97% [80], and the representative sequence of each OTU was aligned by UPARSE [81]. The taxonomy of the representative sequences was classified using the RDP classifier algorithm [82] against the Bacterial 16S database and UNITE Fungal ITS trainset 7 April 2014 database for bacteria and fungi, respectively, with a confidence cutoff of 80%. The relative abundance of a specific OTU was calculated as the number of sequences corresponding to this OTU divided by the total number of sequences of all OTUs of the sample.

Variance partitioning analysis (VPA) was performed with the “vegan” package (Version 2.5-7) in R (Version 4.0.2) to assess the effect of film mulching, soil microbiota and the interaction of these factors on the growth of soybeans [83]. The alpha diversity of the Shannon index and the beta diversity of principal coordinate analysis (PCoA) based on Bray-Curtis distance were implemented and plotted with the package “vegan” (Version 2.5-7) and the package “ggplot2” (Version 3.3.6) in R (Version 4.0.2) [84,85] to estimate the differences in diversity and structure of microbial communities in the two treatments. Linear discriminant analysis of effect size (LEfSe) was applied online (http://huttenhower.sph.harvard.edu/galaxy/ (accessed on 20 September 2021)) at the OTU level to identify the significant (LDA > 2) differences between the two treatments in bacteria and fungi [86]. The fold change of each OTU with plastic film (CF) relative to those without plastic film (C) was calculated with the package “DESeq2” (Version 1.30.1) in R (Version 4.0.2) [87,88]. The random forest model was used to forecast the significant contribution of rhizosphere microbiota to soybean yield with the package “rfPermute” (Version 2.2) in R (Version 4.0.2) [89]. Linear regression analyses relating the relative abundance of rhizosphere microbes to soybean yield were conducted. Spearman’s rank correlation coefficients, false discovery rate (FDR) correction *p*-values [90], and heatmap analysis were utilized to evaluate the relationship of bulk soil characteristics linked to potential key rhizosphere microbes with the package “pheatmap” (Version 1.0.12) in R (Version 4.0.2) [91].

Student’s *t*-test and two-way analysis of variance (Two-way ANOVA) were conducted to assess the significant (*p* < 0.05) differences in the data. Permutational multivariate analysis of variance (PERMANOVA) was carried out to evaluate the effect of treatments on the structure of microbial communities [92].

## 5. Conclusions

In conclusion, the application of plastic film mulching on saline-alkali fields modified the soil properties, resulting in the reduction in soil salinity and the maintenance of soil moisture. The ameliorated soil characteristics as drivers altered the rhizosphere bacterial and fungal community structures, which were predicted to be the major factors contributing to yield. These changes mainly reflected the differences in the taxa and abundance of bacteria and fungi that were enriched in the soybean rhizosphere soil. The potential key rhizosphere microbes were detected markedly higher abundance and 100% enrichment in the CF treatment when compared with the C treatment, resulting in an increase in soybean yield. However, the growth-promoting effect of the potential key rhizosphere microbes remains theoretical in our study. Our subsequent work mainly focuses on sieving and validating the functional microbes and emphasizes the interactions between soybean and rhizosphere microbiota by connecting to rhizosphere metabolites.

## Figures and Tables

**Figure 1 plants-12-01889-f001:**
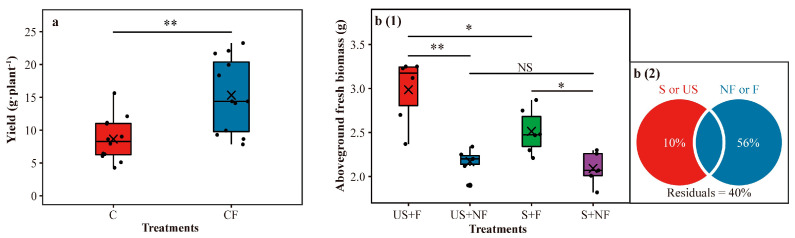
Yield of soybean of field survey (**a**). Aboveground fresh biomass of soybean (TZX-805) in the greenhouse experiment (**b** (1)). Variance partitioning analysis (VPA) of aboveground fresh biomass in the greenhouse experiment (**b** (2)). In panel (**a**), C = chemical fertilizer, CF = chemical fertilizer with plastic film mulching. In panel (**b** (1)), US + F = unsterilized soil with plastic film mulching, US + NF = unsterilized soil without plastic film mulching, S + F = sterilized soil with plastic film mulching, and S + NF = sterilized soil without plastic film mulching. In all panels, asterisks above the boxes indicate significant differences between the two treatments as determined by Student’s *t* test using 0.05 as the boundary (** = *p* < 0.01, * = *p* < 0.05, NS = not significant). The crosses within boxes represent averages. The horizontal lines at the top, middle and bottom of the boxes represent the value at the 75%, 50% and 25% position of the data, respectively. The upper and lower vertical lines of the boxes span no more than 1.5 times the interquartile range.

**Figure 2 plants-12-01889-f002:**
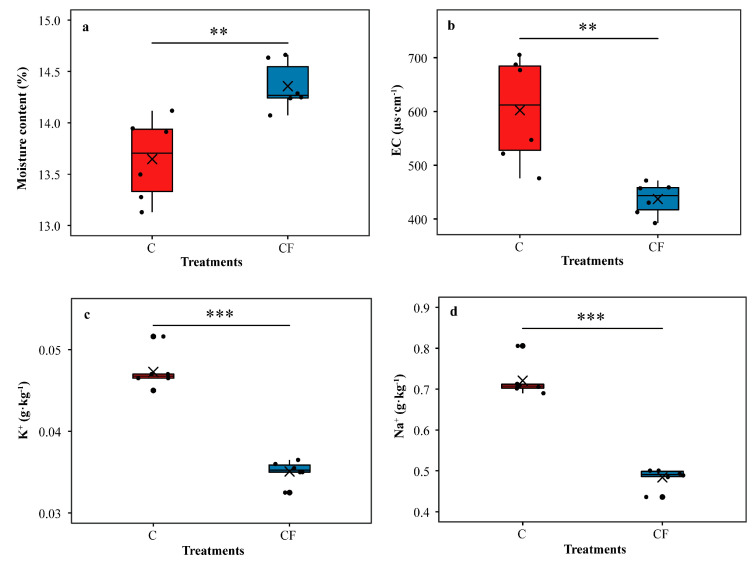
Characteristics of bulk soil of field survey. Moisture content (**a**), EC (electrical conductivity) (**b**), K^+^ (**c**) and Na^+^ (**d**). C = chemical fertilizer, CF = chemical fertilizer with plastic film mulching. Asterisks above the boxes indicate significant differences between the two treatments as determined by Student’s *t* test using 0.05 as the boundary (*** = *p* < 0.001, ** = *p* < 0.01). The crosses within boxes represent averages. The horizontal lines at the top, middle and bottom of the boxes represent the value at the 75%, 50% and 25% position of the data, respectively. The upper and lower vertical lines of the boxes span no more than 1.5 times the interquartile range.

**Figure 3 plants-12-01889-f003:**
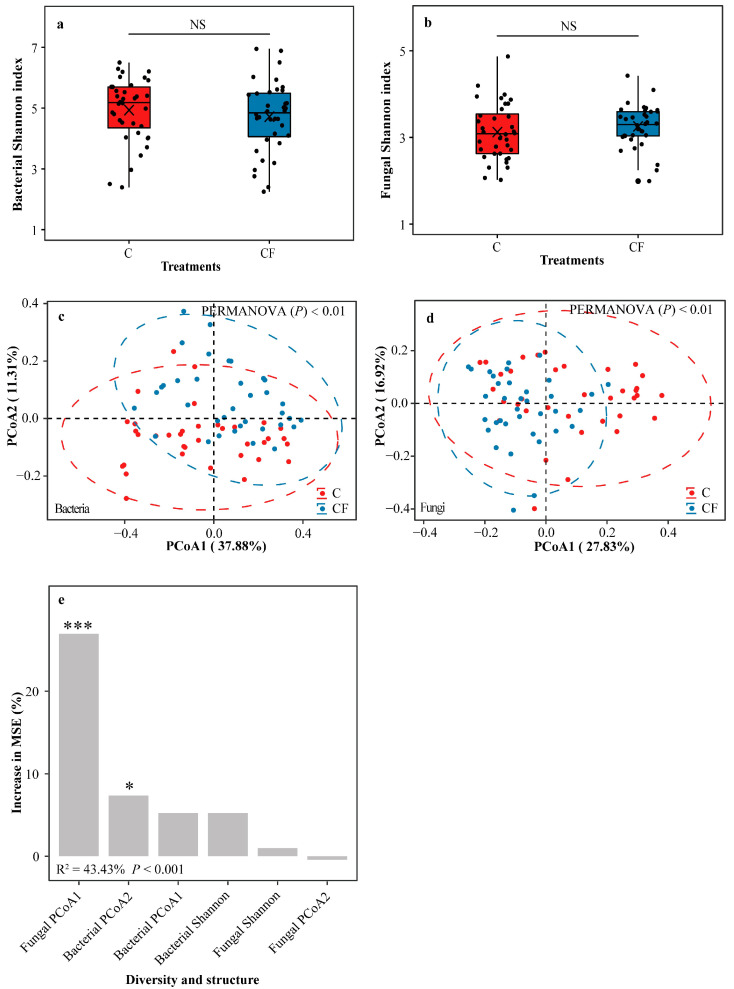
Shannon index of bacterial (**a**) and fungal (**b**) community diversities. Principal coordinate analysis (PCoA) of bacterial (**c**) and fungal (**d**) community structures based on Bray-Curtis distance. Random forest mean predictor importance of bacterial and fungal community diversities and structures for soybean yield (**e**). C = chemical fertilizer, CF = chemical fertilizer with plastic film mulching. In panel (**a**,**b**), markers above the boxes indicate significant differences between the two treatments as determined by Student’s *t* test using 0.05 as the boundary (NS = not significant). The crosses within boxes represent averages. The horizontal lines at the top, middle and bottom of the boxes represent the value at the 75%, 50% and 25% position of the data, respectively. The upper and lower vertical lines of the boxes span no more than 1.5 times the interquartile range. In panel (**c**,**d**), ellipses cover 95% of the data for each treatment. Results of permutational multivariate analysis of variance (PERMANOVA) indicate significant differences of bacterial and fungal community structures in the two treatments. In panel (**e**), percentage increases in the MSE (mean squared error) of predictors represent the important role of considered factors, and higher MSE% value means more important the factor (*** = *p* < 0.001, * = *p* < 0.05).

**Figure 4 plants-12-01889-f004:**
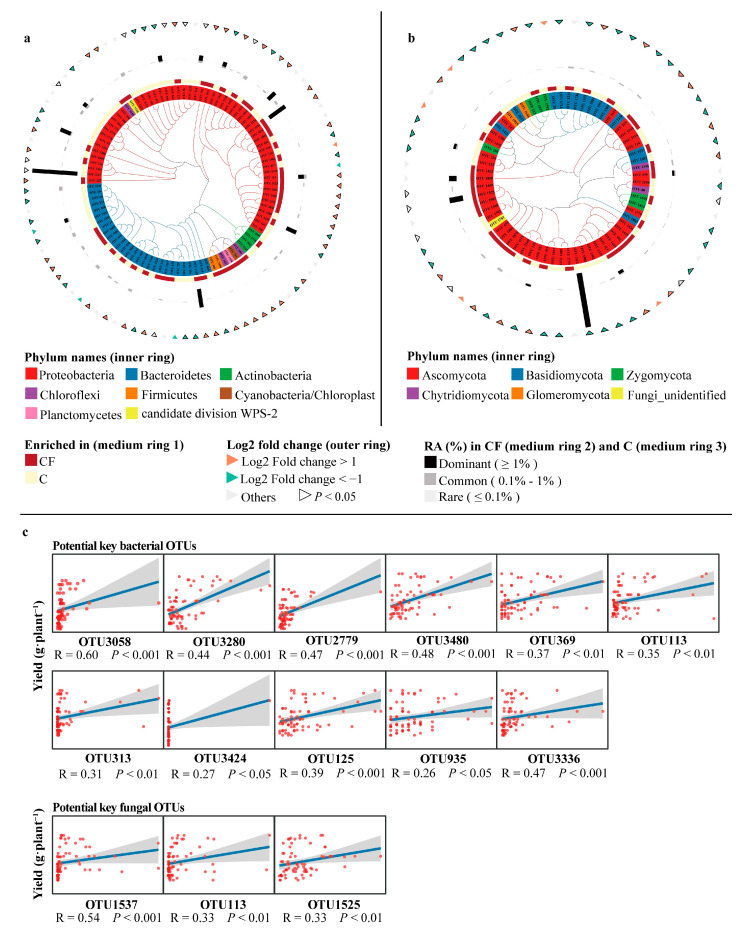
Cladogram showing the phylogenetic relationships among 95 rhizosphere bacterial OTUs (**a**) and 65 rhizosphere fungal OTUs (**b**). Scatter plots illustrating the correlation between soybean yield and the relative abundance of potential key rhizosphere bacterial and fungal OTUs (**c**). C = chemical fertilizer, CF = chemical fertilizer with plastic film mulching. OTUs, operational taxonomic units. RA, relative abundance. In panel (**a**,**b**), leaf labels indicate differential OTU IDs in the two treatments. Rings, from inner to outer, represent: (1) phylum-level taxonomy of OTUs; (2) significant (LDA > 2) OTUs between the two treatments; (3) OTU relative abundance in the CF treatment; (4) OTU relative abundance in the C treatment; and (5) fold change of OTUs. In panel (**c**), *p*-values were evaluated, and significant correlations were determined at *p* < 0.05.

**Figure 5 plants-12-01889-f005:**
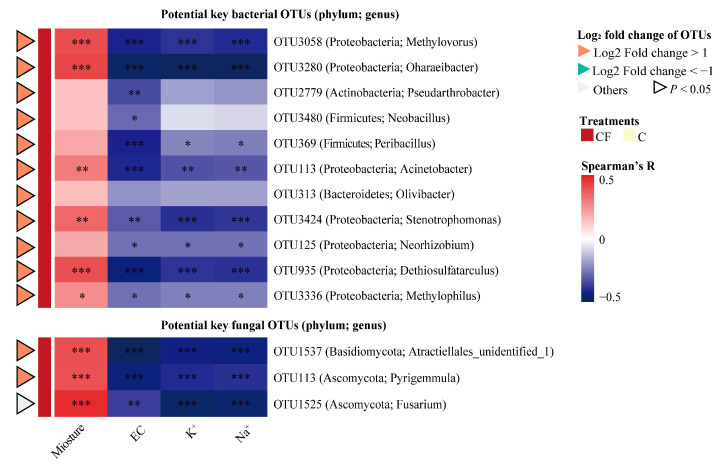
Heatmap illustrating the effects of bulk soil characteristics as drivers on the relative abundance of potential key rhizosphere bacterial and fungal OTUs. C = chemical fertilizer, CF = chemical fertilizer with plastic film mulching. OTUs, operational taxonomic units. The color keys relate the heatmap colors to Spearman’s rank correlation coefficient. *** = *p* < 0.001, ** = *p* < 0.01, * = *p* < 0.05.

**Figure 6 plants-12-01889-f006:**
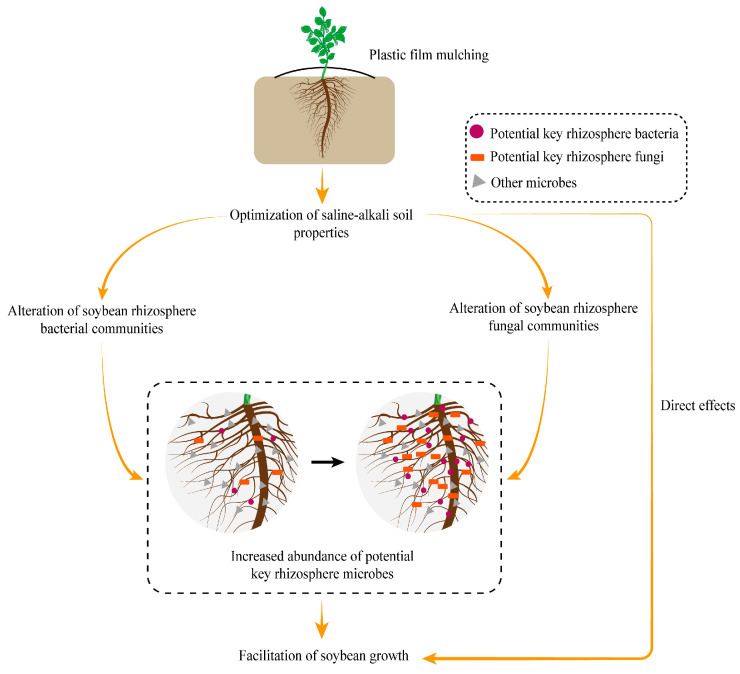
Conceptual model displaying the mechanisms of plastic film mulching promoting the growth of soybeans in saline–alkali soil.

**Table 1 plants-12-01889-t001:** Spearman’s rank correlation results between yield and potential key rhizosphere microbial OTUs.

	OTU IDs	Genus	Spearman’s Rank Correlation	
			*P*	R
Potential key rhizosphere bacterial OTUs	OTU3058	*Methylovorus*	2.34 × 10^−8^	0.60
	OTU3280	*Oharaeibacter*	1.13 × 10^−4^	0.44
	OTU2779	*Pseudarthrobacter*	3.47 × 10^−5^	0.47
	OTU3480	*Neobacillus*	1.94 × 10^−5^	0.48
	OTU369	*Peribacillus*	1.35 × 10^−3^	0.37
	OTU113	*Acinetobacter*	2.29 × 10^−3^	0.35
	OTU313	*Olivibacter*	7.34 × 10^−3^	0.31
	OTU3424	*Stenotrophomonas*	2.30 × 10^−2^	0.27
	OTU125	*Neorhizobium*	7.31 × 10^−4^	0.39
	OTU935	*Dethiosulfatarculus*	2.91 × 10^−2^	0.26
	OTU3336	*Methylophilus*	3.60 × 10^−5^	0.47
Potential key rhizosphere fungal OTUs	OTU1537	*Atractiellales_unidentified_1*	1.01 × 10^−6^	0.54
	OTU113	*Pyrigemmula*	4.65 × 10^−3^	0.33
	OTU1525	*Fusarium*	4.54 × 10^−3^	0.33

Note: OTUs, operational taxonomic units.

## Data Availability

The raw sequence data supporting these findings were submitted to the NCBI Sequence Read Archive (SRA) database (http://www.ncbi.nlm.nih.gov/ (accessed on 3 November 2022)), under BioProject accession number PRJNA897718.

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
