# Peer review of "Rhizosphere Microbiota Promotes the Growth of Soybeans in a Saline–Alkali Environment under Plastic Film Mulching"

_plants, 2023, doi:10.3390/plants12091889_

Round 1

Reviewer 1 Report

The article is interesting, but it needs some improvements:

Brief information related to the research methodology must be entered in the Abstract and please include a few special quantitative achievements from the findings of the study by combining the research objectives and the problems. 

Please include the objectives and novelty in the last paragraph of the introduction section.

At Results, in tables, all abbreviations must be explained in footnotes. 

The study is complex, in several directions and quite difficult to follow, perhaps a schematic representation of the results would be helpful.

 Discussions must be separated from conclusions in this case, and the conclusions must be presented in such a way as to be the basis of a new research direction, especially in the context in which the research does not completely clarify the influence  of plastic film mulching on bacteria and fungi in the soil.

Reviewer 2 Report

#commentary and observations

The themes of the work take into account contemporary problems of soil salinity and pH.

Note the very high importance of the rhizosphere microbiota in plant-microbiota and microbiota-plant interactions.

The authors correctly selected the methodology of their research, to which I have no objections.

The statistical methods are correct.

The Results and Discussion chapters are described and graphically presented in a very readable manner.

The selection of literature and its citation is not objectionable.

From the reviewer's position, I believe that the Abstract section should contain more information regarding the research methodology and include the main results.

In my opinion, the work can be published after these minimal corrections
